# Magnetic Monopoles and Superinsulation in Josephson Junction Arrays

**Carlo Trugenberger [1], M. Cristina Diamantini [2], Nicola Poccia [3], Flavio S. Nogueira [4] and Valerii M. Vinokur [5,6,*]**

[1] SwissScientific Technologies SA, rue du Rhone 59, CH-1204 Geneva, Switzerland; ca.trugenberger@bluewin.ch

[2] NiPS Laboratory, INFN and Dipartimento di Fisica e Geologia, University of Perugia, via A. Pascoli, I-06100 Perugia, Italy; cristina.diamantini@pg.infn.it

[3] Institute for Metallic Materials, Leibniz IFW Dresden, Helmholtzstraße 20, 01069 Dresden, Germany; n.poccia@ifw-dresden.de

[4] Institute for Theoretical Solid State Physics, IFW Dresden, Helmholtzstraße 20, 01069 Dresden, Germany; f.de.souza.nogueira@ifw-dresden.de

[5] Materials Science Division, Argonne National Laboratory, 9700 S. Cass Avenue, Argonne, IL 60637, USA

[6] Consortium for Advanced Science and Engineering (CASE), University of Chicago, 5801 S Ellis Ave, Chicago, IL 60637, USA

\* Correspondence: vinokur@anl.gov

**Abstract:** Electric-magnetic duality or S-duality, extending the symmetry of Maxwell's equations by including the symmetry between Noether electric charges and topological magnetic monopoles, is one of the most fundamental concepts of modern physics. In two-dimensional systems harboring Cooper pairs, S-duality manifests in the emergence of superinsulation, a state dual to superconductivity, which exhibits an infinite resistance at finite temperatures. The mechanism behind this infinite resistance is the linear charge confinement by a magnetic monopole plasma. This plasma constricts electric field lines connecting the charge–anti-charge pairs into electric strings, in analogy to quarks within hadrons. However, the origin of the monopole plasma remains an open question. Here, we consider a two-dimensional Josephson junction array (JJA) and reveal that the magnetic monopole plasma arises as quantum instantons, thus establishing the underlying mechanism of superinsulation as two-dimensional quantum tunneling events. We calculate the string tension and the dimension of an electric pion determining the minimal size of a system capable of hosting superinsulation. Our findings pave the way for study of fundamental S-duality in desktop experiments on JJA and superconducting films.

**Keywords:** S-duality; magnetic monopoles; superinsulation; Josephson junction arrays; electric strings; instantons

## 1. Introduction

The superinsulating state, dual to superconductivity [1–7], is a remarkable manifestation of S-duality [8] in condensed matter physics. Superinsulators exhibit infinite resistance at finite temperatures, mirroring the infinite conductance of superconductors. The mechanism preventing charge transport is the linear charge confinement [7] of both Cooper pairs and normal excitations by a magnetic monopole plasma. This plasma constricts electric field lines connecting the charge–anti-charge pairs into electric strings, in analogy to quarks within hadrons [9].

Maxwell equations in vacuum are symmetric under the duality transformations interchanging the electric and magnetic fields $\mathbf{E} \to \mathbf{B}$ and $\mathbf{B} \to -\mathbf{E}$ (we use hereafter natural units $c = 1$, $\hbar = 1$, and $\varepsilon_0 = 1$).

This duality holds in the presence of field sources, provided magnetic monopoles [8] are included along with electric charges. While, despite intensive searches [10], no elementary particles with a net magnetic charge have ever been observed, monopoles emerge and are detected as topological excitations in strongly correlated systems (see, for example [11,12]). Notably, these monopoles emerge as classical particles that freeze out upon cooling down the system. A drastically different class of phenomena arises if monopoles form a monopole plasma as a result of multiple instanton quantum tunneling events. In this case, a monopole plasma offers an ideal screening mechanism for electric fields, and the system harboring the monopole plasma makes a perfect dielectric with zero static dielectric constant, $\varepsilon = 0$, as long as the electric field does not exceed some threshold value [13]. Now, as the perfect diamagnetism is associated with an infinite conductance, i.e., superconductivity, the perfect dielectricity should correspond to dual superconductors possessing an infinite resistance, i.e., superinsulators [1,5].

Dual superconductivity was introduced in the 1970s by 't Hooft as a Gedankenexperiment for quark confinement [9]. The idea was exactly that perfect dielectricity, in analogy to the Meissner effect in superconductors, would squeeze electric fields into thin flux tubes with quarks at their ends. When quarks are pulled apart, it is energetically more favorable to pull out of the vacuum additional quark–antiquark pairs and to form several short strings instead of a long one. As a consequence, the color charge can never be observed at distances above the fundamental length scale, $1/\Lambda_{QCD}$, and quarks are confined. Only color-neutral hadron jets can be observed in collider events.

Superinsulation as an emergent condensed matter state was first proposed in [1] on the basis of electric-magnetic duality and independently reinvented in [5] on the basis of duality between two different symmetry realizations of the uncertainty principle. Experiments reporting superinsulation detected it in films experiencing superconductor-insulator transition (SIT) [3,4]. Both considerations [1,5] involved the symmetric interchange of charges and vortices in 2D systems, Finally, the topological gauge theory of superinsulation put forth in [7] revealed that the relevant fundamental duality is the one relating charges and magnetic monopoles rather than vortices. Accordingly, superinsulation is the result of the proliferation of the monopole plasma and represents the Abelian realization of dual superconductivity [14] in condensed matter. The experimental implications, including the Berezinskii–Kosterlitz–Thouless (BKT) criticality of the deconfinement transition and the electric field-induced breakdown of confinement, were observed in NbTiN films [13,15]. However, the monopoles introduced in [7] have emerged in the framework of a long-distance effective field theory of thin films. Here, we complete the description of superinsulation and consider a Josephson junction array (JJA), which, in particular, represents a "microscopic" model for a superconducting film [16], and develop an exact magnetic monopole theory of superinsulation in JJA.

## 2. Results

We start with the notion that, contrary to charges, vortices are topological excitations, characterized by a topological quantum number. The configuration space of the theory of vortices decomposes into so-called superselection sectors, characterized by the integer total vortex number, which are connected via instantons, non-perturbative configurations representing quantum tunneling events between topological vacua [17]. As a consequence, charges are conserved but vortices are not and can "appear" and "disappear" via quantum tunneling events forming the instantons. In two spatial dimensions (2D), these instantons are nothing but magnetic monopoles [18]. The instantons are known to make a noticeable impact on the low-temperature physics of one-dimensional (1D) system. In particular, the global O(2) model, representing the physics of 1D superconducting quantum wires with screened Coulomb interactions, admits instantons representing quantum phase slips [18]. These quantum phase slips cause a superconductor-to-metal quantum transition [19,20] at zero temperature, an insulating phase possibly emerging in finite systems coupled to the environment [21]. Remarkably, in 2D [7,13], the monopole instantons manifest a much more profound and striking action, governing not only metallic but superinsulating behavior.

We consider a square Josephson junction array (JJA) with the spacing $\ell$ comprising superconducting islands with the nearest-neighbor Josephson coupling of the strength $E_J$. Each island has a self-capacitance $C_0$ and mutual capacitances $C$ to its nearest neighbors. The corresponding charging energies are $E_{C_0} = e^2/2C_0$ and $E_C = e^2/2C$. The degrees of freedom of the array are the integer multiples of the fundamental charge unit $2e$ of the Cooper pair on each island, $q_{\mathbf{x}} \in \mathbb{Z}$, and the quantum-mechanically conjugated phases $\varphi_{\mathbf{x}} \in [0, 2\pi]$. The partition function for such a JJA [16] is given by (see Section 4).

$$Z = \sum_{\{q\}} \int_{-\pi}^{+\pi} \mathcal{D}\varphi \exp(-S) ,$$
$$S = \sum_x i\, q_x \Delta_0 \varphi_x + 4\ell_0 E_C\, q_x \frac{1}{C_0/C - \Delta} q_x + \sum_{x,i} \ell_0 E_J \left(1 - \cos\left(\Delta_i \varphi_x\right)\right) , \tag{1}$$

where $S$ is the Euclidean action and the sum runs over the 3D Euclidean lattice with spacing $\ell_0$ in the "time" direction, which, as we show below, represents the (inverse) tunneling frequency. Here, $\Delta_i$ and $\hat{\Delta}_i$ are forward and backward finite differences, $\Delta \equiv \hat{\Delta}_i \Delta_i$ is the corresponding 2D finite difference Laplacian, and $\Delta_0$ and $\hat{\Delta}_0$ are forward and backward finite time differences (see Section 4). The integer charges $q_{\mathbf{x}}$ interact via the two-dimensional Yukawa potential with the mass $\sqrt{C_0/C}/\ell$. In the experimentally accessible nearest-neighbors capacitance limit $C \gg C_0$, this implies a two-dimensional Coulomb law at distances smaller than the electrostatic screening length $\Lambda = \ell \sqrt{C/C_0}$. Then, the charging energy $E_C$ and the Josephson coupling $E_J$ are the two relevant energy scales which can be further traded for one energy parameter $\omega_P = \sqrt{8E_C E_J}$, the Josephson plasma frequency, and one numerical parameter $g = \sqrt{\pi^2 E_J/2E_C}$, the dimensionless conductance. In the following, we consider the physics of JJA at energies much below the plasma frequency, which takes the role of the natural ultraviolet (UV) cutoff in the theory.

In the limit $C_0 = 0$, which we henceforth consider, the partition function of the JJA can be mapped exactly [1] onto the partition function of a topological Chern–Simons gauge theory [22] (see Section 4),

$$Z = \sum_{\{Q_i\}} \sum_{\{M_\mu\}} \int \mathcal{D}a_\mu \mathcal{D}b_\mu \exp(-S) ,$$
$$S = \sum_x i2\pi\, a_\mu K_{\mu\nu} b_\nu + \frac{1}{2\ell_0 E_J} j_i^2 + \frac{\pi^2}{4\ell_0 E_C} \phi_i^2 + i2\pi a_i Q_i + i2\pi b_\mu M_\mu , \tag{2}$$

where $K_{\mu\nu}$ is the lattice Chern–Simons operator [1] (see Section 4). Here, $a_\mu$ and $b_\mu$ are fictitious gauge fields representing conserved charge and vortex fluctuations by their dual field strengths, $j_\mu = K_{\mu\nu} b_\nu$ and $\phi_\mu = K_{\mu\nu} a_\nu$, respectively. The first term in the action is the topological mixed Chern–Simons term [22] between these two types of dual fluctuations. The integers $Q_i$ are the electric topological excitations of the system, the integers $M_i$ are the magnetic topological excitations. Together with the vortex number $M_0$, the latter form a three-current $M_\mu$ which is conserved due to the gauge invariance in the $b_\mu$ gauge sector, $\hat{\Delta}_\mu M_\mu = 0$. Due to this constraint, only the two integers $M_i$ are the independent degrees of freedom. From the point of view of the original Minkowski space-time, the three-current $M_\mu$ describes events in which one vortex disappears from the array, the flux being "carried away" by the spatial vortex currents $M_i$. From the Euclidean space-time point of view, however, $M_\mu$ are the components of a 3D magnetic field. A configuration such as the one in Figure 1 thus represents a unit magnetic monopole, the JJA vortex on the lower plaquette playing the role of the Dirac string [8]. The integer monopole charge is $m = \Delta_i M_i$. The asymmetry of the monopole, whose flux flows out only in the spatial directions, but not over a whole 3D lattice cube, is due to the deep non-relativistic limit of the JJA gauge theory. One sees in Figure 1 how the flux of the JJA vortex is divided up into four parts and is carried away by the $M_i$ in the spatial directions. As a consequence, on the upper plaquette of the cube, representing the same JJA plaquette one quantum of time later, there is no longer any vortex. Thus, the magnetic monopole $m$ expresses the tunneling of the system between two different topological vacua.

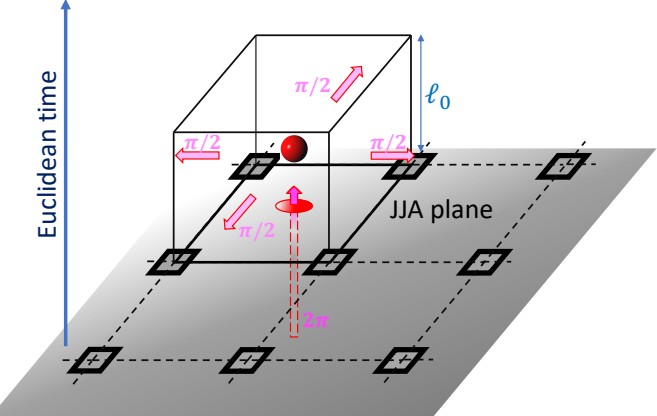

**Figure 1. Magnetic monopole in Josephson junction array.** A magnetic monopole instanton depicted by the red ball is assigned to the center of the 3D cube in the 3D Euclidean lattice with the spatial spacing $\ell$, comprising a single JJA plaquette and the elemental unit $\ell_0$ along the quantized Euclidean time. An elemental fluxon carrying the phase $2\pi$ of a single vortex located in the JJA plaquette splits into four parts each carried away through the vertical plaquettes, and the original JJA plaquette one unit time later does not contain the vortex anymore. The monopole tunneling event interpolates between two states differing by one unit of the topological quantum number. The quantity $\ell_0\ell^2$ can be considered as the volume of the monopole. The monopole itself is anisotropic, with no flux coming out in the third direction, because of the deep non-relativistic limit of the effective compact QED action in JJA.

We now discuss the implications of the monopole plasma proliferation. This occurs in the phase where the electric topological excitations $Q_i$ are suppressed because of their large energy, and one sets $Q_i = 0$. To establish the nature of the monopole plasma phase, we derive the electromagnetic response of the system by coupling the charge current $j_\mu = K_{\mu\nu}b_\nu$ to the real physical electromagnetic potential $A_\mu$

$$S \to S + i\sum_x A_\mu j_\mu = S + i\sum_x A_\mu K_{\mu\nu} b_\nu \,, \tag{3}$$

where $S$ is the Chern–Simons gauge theory action introduced in Equation (2). Integrating out the matter fields, and taking the limit $\ell_0\omega_P \gg 1$, one arrives at the effective action for electromagnetic fields in the monopole plasma phase as

$$S\left(A_\mu, M_i\right) = \frac{g}{4\pi\ell_0\omega_P} \sum_x (F_i - 2\pi M_i)^2 \,, \tag{4}$$

where $F_i$ are the spatial components of the dual electromagnetic vector strength $F_\mu = \hat{K}_{\mu\nu}A_\nu$ and the magnetic topological excitations, encoding the monopoles, are now additional dynamical variables which have to be summed over in the partition function. This is a deep non-relativistic version of Polyakov's compact QED action [14,18], in which only electric fields survive. Its form shows that the action is periodic under shifts $F_i \to F_i + 2\pi M_i$, with integer $M_i$ and that the gauge fields are thus indeed compact, i.e., angular variables defined on the interval $[-\pi, +\pi]$.

One is used to the fact that electromagnetic fields mediate Coulomb forces between static charges, a $1/|\mathbf{x}|$ potential in 3D, or a $\log|\mathbf{x}|$ potential in 2D. Monopoles in compact electromagnetism drastically change this, as we now show. We consider two external probe charges of strength $\pm q_{\text{ext}}$ and compute how their interaction potential is changed by the monopoles. To do so, we consider the expectation value of the Wilson loop operator $W(C)$, where $C$ is a closed loop in 3D Euclidean space-time (a factor $\ell$ is absorbed into the gauge field $A_\mu$ to make it dimensionless),

$$\langle W(C)\rangle = \frac{1}{Z_{A_\mu, M_i}} \sum_{\{M_i\}} \int_{-\pi}^{+\pi} \mathcal{D}A_\mu \, e^{-\frac{g}{4\pi\ell_0\omega_P} \sum_x (F_i - 2\pi M_i)^2} e^{iq_{\text{ext}}\sum_C A_\mu} \,. \tag{5}$$

When the loop $C$ is restricted to the plane formed by the Euclidean time and one of the space coordinates, $\langle W(C) \rangle$ measures the potential between two external probe charges $\pm q_{\text{ext}}$. A perimeter law indicates a short-range potential, while an area-law is tantamount to a linear interaction between the probe charges [17,18].

For couplings $g/\ell_0 \omega_{\text{P}}$ large enough, the action peaks near $F_i = M_i$, allowing for the saddle-point approximation to compute the Wilson loop. Using the lattice Stoke's theorem, one rewrites Equation (5) as

$$\langle W(C) \rangle = \frac{1}{Z_{A_\mu, M_i}} \sum_{\{M_i\}} \int_{-\pi}^{+\pi} \mathcal{D}A_\mu \; e^{-\frac{g}{4\pi\ell_0\omega_{\text{P}}} \sum_x (F_i - 2\pi M_i)^2} e^{iq_{\text{ext}} \sum_S S_i (F_i - 2\pi M_i)} \,, \tag{6}$$

where the quantities $S_i$ are unit vectors perpendicular to the plaquettes forming the surface $S$ encircled by the loop $C$ and vanish on all other plaquettes. We also multiply the Wilson loop operator by 1 in the form $\exp(-i2\pi q_{\text{ext}} M_i)$ on all plaquettes forming $S$. Following Polyakov [14,18], we decompose $M_i$ into transverse and longitudinal components, $M_i = M^{\text{T}}_i + M^{\text{L}}_i$ with $M^{\text{T}}_i = \epsilon_{ij}\Delta_j n + \epsilon_{ij}\Delta_j \xi$, $M^{\text{L}}_i = \Delta_i \lambda$, where $\{n\}$ are integers and $\Delta\lambda = \hat{\Delta}_i \Delta_i \lambda = m$. The two sets of integers $\{M_i\}$ are thus traded for one set of integers $\{n\}$ and one set of integers $\{m\}$ representing the magnetic monopoles. The integers $\{n\}$ are used to shift the integration domain for the gauge field $A_\mu$ to $[-\infty, +\infty]$. The real variables $\{\xi\}$ are then also absorbed into the gauge field. The integral over this non-compact gauge field $A_\mu$ gives then the Gaussian fluctuations around the instantons $m$, representing the saddle points of the action. Gaussian fluctuations do not contribute to confinement and can thus can be neglected. Only the summation over instantons, $\{m\}$, remains:

$$\langle W(C) \rangle = \frac{1}{Z_{\text{m}}} \sum_{\{m\}} e^{-\frac{\pi g}{\ell_0 \omega_{\text{P}}} \sum_x m_x \frac{1}{-\Delta} m_x} e^{i2\pi q_{\text{ext}} \sum_S \hat{\Delta}_i S_i \frac{1}{-\Delta} m_x} \,. \tag{7}$$

For $q_{\text{ext}} = 1$, i.e., Cooper pair probes, the result is (see Section 4)

$$\langle W(C) \rangle = e^{-\sigma A} \,, \tag{8}$$

where $A$ is the area of the surface $S$ enclosed by the loop $C$. This area law indicates a linear potential between test Cooper pairs, with the string tension

$$\sigma = \frac{\hbar \omega_{\text{P}}}{\ell} \sqrt{\frac{16}{\pi g \ell_0 \omega_{\text{P}}}} \sqrt{z} = \frac{\hbar \omega_{\text{P}}}{\ell} \sqrt{\frac{16}{\pi g \ell_0 \omega_{\text{P}}}} \; e^{-\frac{\pi g}{2\ell_0 \omega_{\text{P}}} G(0)} \,, \tag{9}$$

where $z$ is the instanton fugacity, $G(0)$ is the value of the infrared-regularized 2D lattice Coulomb potential at coinciding points and we reinstate physical units. The string binds together charges, prevents charge transport on arrays of a sufficient size, and is the origin of the infinite resistance characterizing superinsulation. For single electron probes, $q_{\text{ext}} = 1/2$ in our units, the string tension is

$$\sigma_{\text{electrons}} = \frac{1}{2}\sigma \,. \tag{10}$$

Single electrons are thus also confined, hence the absence of charge transport mediated by thermally excited normal quasiparticles in the superinsulating state, which has remained a tantalizing puzzle ever since the experimental discovery of the superinsulation [5].

## 3. Discussion

We are now equipped to address another puzzle of superinsulation—why it was experimentally observed only in films but never in JJA. To resolve it, let us estimate the typical size of the string that confines the charges. Note that the string tension comprises two factors. The first, $\hbar \omega_{\text{P}}/\ell$, depends

solely on the "classical" array parameters, the lattice spacing and plasma frequency. The second factor depends on the quantum characteristics of the array, the dimensionless conductance and the ratio between the long tunneling time $\ell_0$ and the short phase oscillation period $1/\omega_{\mathrm{P}}$. To estimate the typical string size $\ell_{\mathrm{string}} = \sqrt{c\hbar/\sigma}$ we take the typical values of experimental JJA [23], $\ell = 100$ nm and $\omega_{\mathrm{P}} = 10$ GHz. Then, the first contribution to the string size amounts to $\ell^0_{\mathrm{string}}/\ell = \sqrt{c/\ell\omega_{\mathrm{P}}} \approx 550$. This number is reduced by the second factor in (9). However, even for $\ell_0\omega_{\mathrm{P}} = \mathcal{O}(1000)$, we still have $\ell_{\mathrm{string}}/\ell \approx 150$, at the border of the total size, 190, of typical JJA showing the SIT [23]. In field theory parlance, these JJA are too far from the infrared confining fixed point [24] and due to asymptotic freedom (for a review see [17]) only the screened Coulomb forces within an electric "meson" can be observed. To detect superinsulation on JJA one must thus design an array with sufficiently high plasma frequency, with a linear size sufficiently large to fit an entire string, presumably in the thousands of lattice spacings, and, finally, the large ratio $\Lambda/\ell = C/C_0 \gg 1$. As mentioned above, the latter condition implements experimentally our starting model assumption $C_0 \to 0$ and ensures a proper 2D Coulomb interaction between the charges. In JJA, $\Lambda/\ell \simeq 20 \div 40$ at best. This is insufficient to ensure a 2D Coulomb interaction, as evidenced by the absence of the charge BKT transition at the insulating side of the SIT [25], hence no superinsulation. To compare, in films, $\Lambda/\ell \simeq 500 \div 10^5$ [6,15] (in films the UV cutoff $\ell \simeq (3.5 \div 4)\xi$). Now, one immediately can see that since in the JJA of [23,25], $\Lambda \simeq a(\ell/d)$, where $a$ is the lateral size of the individual junction and $d$ is the spacing between the junction electrodes, the attempt to increase $\Lambda$ by, say, a factor of 10 playing with the size of a single junction, would increase $E_{\mathrm{J}}$ by a factor of 100 and reduce $E_{\mathrm{C}}$ by the same factor, which would immediately move the JJA well into the superconducting domain (since we consider a system in the vicinity of the SIT, where $E_{\mathrm{J}} \simeq E_{\mathrm{C}}$), hence the observation of superinsulation in a standard classical JJA at present is hardly possible. A promising platform for highly controllable JJA, capable of hosting superinsulation, is offered by proximity arrays that can be driven through the SIT by either a gate voltage [26] or a magnetic field [27]. Note that, in the system adopted in [27], the dual twin to a Cooper pair Mott insulator, the vortex Mott insulator, has already been observed at the superconducting side of the SIT.

Finally, a comment about the role of disorder is due. In Ref. [28], the origin of superinsulation was related solely to disorder localizing charges. This mix up emerged since, in [28], superinsulation was confused with many-body-localization, which was introduced in the seminal paper [29]. Our results conclusively show that the origin of superinsulation lies in the proliferation of quantum tunneling events (magnetic monopole instantons), which can be viewed as the 2D generalization of 1D quantum slips in wires, with no role of disorder involved.

To conclude, we demonstrated that magnetic monopoles appearing in JJA are a deep non-relativistic version of the monopoles introduced by Polyakov in the framework of compact QED [14,18] and derived how these instantons dominate the JJA dynamics at low energies, far below the JJA plasma frequency. The tension of the string binding charges into neutral "mesons" is expressed through JJA parameters, the distance between superconducting islands, $\ell$, the plasma frequency, $\omega_{\mathrm{P}}$, and the dimensionless conductance $g$. We found that both Cooper pairs and normal excitations are confined by monopoles, thereby resolving the enigma of the absence of current due to single-charge excitations in superinsulators. One of the experimental implications of our results is that the typical JJA used so far are far too coarse and small to accommodate an entire electric string. In field theory parlance, they are too far from the infrared confining fixed point [24] and due to asymptotic freedom [17] only the screened Coulomb forces within an electric "meson" can be observed. The large size of the electric mesons reflects the fact that the electromagnetic interaction is much weaker than the strong force. This explains the paradoxical enigma why superinsulators were experimentally seen in films but not yet in the paradigmatic JJA system for which they were first derived [5,7] and indicates the direction for further experimental research. Devising large-size JJA and proximity arrays will open an opportunity of observing superinsulation in highly controllable and tuneable systems and of exploring the fundamental properties of S-duality via the desktop experiments.

## 4. Materials and Methods

### 4.1. The Model for Josephson Junctions Arrays

The Hamiltonian for a planar JJA of spacing $\ell$, with nearest neighbors Josephson couplings $E_J$, ground capacitances $C_0$, and nearest neighbors capacitances $C$ is given by [16,30]

$$H = \sum_{\mathbf{x}} \frac{C_0}{2} V_{\mathbf{x}}^2 + \sum_{<\mathbf{xy}>} \frac{C}{2} \left(V_{\mathbf{y}} - V_{\mathbf{x}}\right)^2 + E_J \left(1 - \cos\left(\varphi_{\mathbf{y}} - \varphi_{\mathbf{x}}\right)\right), \tag{11}$$

where boldface characters denote the sites of the two-dimensional array, $< \mathbf{xy} >$ indicates nearest neighbors, $V_{\mathbf{x}}$ is the electric potential of the island at $\mathbf{x}$, and $\varphi_{\mathbf{x}}$ is the phase of its order parameter (we use natural units $c = 1$, $\hbar = 1$, and $\varepsilon_0 = 1$). The Hamiltonian (11) can be rewritten as

$$H = \sum_{\mathbf{x}} \frac{1}{2} V_{\mathbf{x}} \left(C_0 - C\Delta\right) V_{\mathbf{x}} + \sum_{\mathbf{x},i} E_J \left(1 - \cos\left(\Delta_i \varphi_{\mathbf{x}}\right)\right), \tag{12}$$

where $\Delta_i$ and $\hat{\Delta}_i$ are forward and backward finite differences and $\Delta \equiv \hat{\Delta}_i \Delta_i$ is the two-dimensional finite difference Laplacian. The phases $\varphi_{\mathbf{x}}$ are quantum-mechanically conjugated to the charges $\mathcal{E}_{\mathbf{x}}$ on the islands: these are quantized in integer multiples of 2e (Cooper pairs), $\mathcal{E}_{\mathbf{x}} = 2eq_{\mathbf{x}}$, $q_{\mathbf{x}} \in Z$, where $e$ is the electron charge. The Hamiltonian (12) can be expressed in terms of charges and phases by noting that the electric potentials $V_{\mathbf{x}}$ are determined by the charges $\mathcal{E}_{\mathbf{x}}$ via a discrete version of Poisson's equation:

$$\left(C_0 - C\Delta\right) V_{\mathbf{x}} = \mathcal{E}_{\mathbf{x}}. \tag{13}$$

Using this in (12), we get

$$H = \sum_{\mathbf{x}} 4E_{\text{C}}\, q_{\mathbf{x}} \frac{1}{C_0/C - \Delta} q_{\mathbf{x}} + \sum_{\mathbf{x},i} E_J \left(1 - \cos\left(\Delta_i \varphi_{\mathbf{x}}\right)\right), \tag{14}$$

where $E_{\text{C}} \equiv e^2/2C$. The integer charges $q_{\mathbf{x}}$ interact via a two-dimensional Yukawa potential of mass $\sqrt{C_0/C}/\ell$. In the nearest-neighbors capacitance limit $C \gg C_0$, which is accessible experimentally, this becomes essentially a two-dimensional Coulomb law.

The partition function of the JJA admits a phase-space path-integral representation [30]

$$\begin{aligned} Z &= \sum_{\{q\}} \int_{-\pi}^{+\pi} \mathcal{D}\varphi \exp(-S), \\ S &= \int_0^\beta dt \sum_{\mathbf{x}} i\, q_{\mathbf{x}} \dot{\varphi}_{\mathbf{x}} + 4E_{\text{C}}\, q_{\mathbf{x}} \frac{1}{C_0/C - \Delta} q_{\mathbf{x}} \\ &\quad + \sum_{\mathbf{x},i} E_J \left(1 - \cos\left(\Delta_i \varphi_{\mathbf{x}}\right)\right), \end{aligned} \tag{15}$$

where $\beta = 1/T$ is the inverse temperature. In (15), time also has to be considered as discrete, as generally appropriate when degrees of freedom can change only in integer steps. We thus introduce a discrete time step $\ell_0$ whose inverse represents the ultraviolet (UV) energy cutoff in the model. The interval $\ell_0$ represents the minimal time interval on which the dynamics is still governed by the horizontal Hamiltonian (14). For frequencies above $1/\ell_0$, new modes can be excited. We thus substitute the time integrals and space sums over a lattice with nodes $\mathbf{x}$ by a sum over space-time lattice nodes $x$, with $x^0 = t$ standing for the discrete time direction. Denoting by $\Delta_0$ the (forward) finite time differences, we obtain Equation (1) of the main text.

### 4.2. Lattice Chern–Simons Operator

Formulating a lattice version of the Chern–Simons operator $\epsilon_{\mu\alpha\nu}\partial_\alpha$ requires some care, if gauge invariance has to be properly implemented [1]. We introduce first the forward and backward finite difference and shift operators on the 3D Euclidean lattice with sites denoted by $\{x\}$ and directions indicated by Greek letters and lattice spacing, $d$:

$$\Delta_\mu f(x) = f(x + d\hat{\mu}) - f(x) , \qquad S_\mu f(x) = f(x + d\hat{\mu}) ,$$
$$\hat{\Delta}_\mu f(x) = f(x) - f(x + d\hat{\mu}) , \qquad \hat{S}_\mu f(x) = f(x - d\hat{\mu}) , \tag{16}$$

where $\hat{\mu}$ denotes a unit vector in direction $\mu$, $d = \ell$ in the spatial directions, and $d = \ell_0$ in the Euclidean time direction. Summation by parts on the lattice interchanges both the two finite differences (with a minus sign) and the two shift operators. Gauge transformations are defined by using the forward finite differences. In terms of these operators, one can then define two lattice Chern–Simons terms

$$K_{\mu\nu} = S_\mu \epsilon_{\mu\alpha\nu} \Delta_\alpha , \qquad \hat{K}_{\mu\nu} = \epsilon_{\mu\alpha\nu} \hat{\Delta}_\alpha \hat{S}_\nu , \tag{17}$$

where no summation is implied over the equal indices $\mu$ and $\nu$. Summation by parts on the lattice interchanges also these two operators (without any minus sign). Gauge invariance is then guaranteed by the relations

$$K_{\mu\alpha} \Delta_\nu = \hat{\Delta}_\mu K_{\alpha\nu} = 0 , \qquad \hat{K}_{\mu\nu} \Delta_\nu = \hat{\Delta}_\mu \hat{K}_{\mu\nu} = 0 . \tag{18}$$

Note that the product of the two Chern–Simons terms gives the lattice Maxwell operator

$$K_{\mu\alpha} \hat{K}_{\alpha\nu} = \hat{K}_{\mu\alpha} K_{\alpha\nu} = -\delta_{\mu\nu} \Delta + \Delta_\mu \hat{\Delta}_\nu , \tag{19}$$

where $\Delta = \hat{\Delta}_\mu \Delta_\mu$ is the 3D Laplace operator.

### 4.3. Deriving the Chern–Simons Gauge Theory

We first use the Villain representation (for a review, see [31]) to express the cosine interaction in (15) in terms of a set of integer link variables $a_i$. By introducing real charge currents $j_i$ and assuming that the size of the system is much smaller than $\Lambda$, so that we can safely set $C_0 \to 0$ from now on, we arrive at

$$Z = \sum_{\{a_i\},\{j_0\}} \int \mathcal{D}j_i \int_{-\pi}^{+\pi} \mathcal{D}\varphi \exp(-S) ,$$
$$S = \sum_x i\, j_0 \Delta_0 \varphi + i j_i \left( \Delta_i \varphi + 2\pi a_i \right) + 4\ell_0 E_C \, j_0 \frac{1}{-\Delta} j_0 + \frac{1}{2\ell_0 E_J} j_i^2 , \tag{20}$$

where we drop the subscripts referring to the lattice positions of the variables and where the summation over equal Greek 3D lattice direction indices is implied. We also introduce the notation $j_0$ for the integer charges.

To proceed further, we stress that Equation (20), derived from what is viewed as the standard JJA Hamiltonian, misses a crucial piece. This missing contribution is a kinetic term proportional to the vortex mass. The omission of this term is common practice when considering overdamped junctions. However, this omission is a priori not justified in arrays, in which collective effects may lead to a renormalization of the vortex mass, no matter how large its bare value may be. It is known since the very early days of JJA that integrating over charge fluctuations leads anyway to a vortex kinetic term [30]. It is a general principle in field theory that whatever is induced by fluctuations *must* be included at bare level. It is also known that dissipation is substantially reduced when the Coulomb interaction becomes long-range [32], exactly the regime we are interested in. Finally, ballistic motion of vortices has indeed been observed experimentally [33]. This calls for the necessity of adding a vortex kinetic term to the action. Such a vortex kinetic term would involve the time derivative of $a_i$, the integer variable conjugated to the charge currents, and would represent phase slips corresponding to one vortex moving from one plaquette to a neighboring one. When the coefficient of this kinetic term $(\partial_0 a_i)^2$ takes the value $\pi^2/4\ell_0 E_C$ we can introduce a real Lagrange multiplier $a_0$ and a fictitious electric field $\phi_i = K_{i\mu} a_\mu$ and write the vortex kinetic term and the charge Coulomb interaction compactly as

$$Z = \sum_{\{a_i\},\{j_0\}} \int \mathcal{D}a_0 \mathcal{D}j_i \int_{-\pi}^{+\pi} \mathcal{D}\varphi \exp(-S) ,$$
$$S = \sum_x i\, j_0 \left( \Delta_0 \varphi + 2\pi a_0 \right) + i j_i \left( \Delta_i \varphi + 2\pi a_i \right) + \frac{1}{2\ell_0 E_J} j_i^2 + \frac{\pi^2}{4\ell_0 E_C} \phi_i^2 . \tag{21}$$

In this representation, the Coulomb interaction between the charges follows from the Gauss law constraint associated with the Lagrange multiplier $a_0$. This procedure works only at a particular value of the vortex mass that makes the model self-dual under the interchange of charge and vortex variables while simultaneously interchanging $\pi^2 E_J \leftrightarrow 2E_C$, or alternatively, $g \leftrightarrow 1/g$. As a consequence, it was called the self-dual approximation in [1]. We adopt this approximation here too.

At this point, we note that the charge current $j_\mu$ is conserved and, hence, it can be represented as the field strength associated to a second fictitious gauge field $b_\mu$ as $j_0 = K_{0i}b_i$, $j_i = K_{i0}b_0 + K_{ij}b_j$, where $b_0$ is a real variable, while $b_i$ are integers. We then use Poisson's formula,

$$\sum_{n_\mu} f\left(n_\mu\right) = \sum_{k_\mu} \int dn_\mu f\left(n_\mu\right) e^{i2\pi n_\mu k_\mu} , \tag{22}$$

turning a sum over integers $\{n_\mu\}$ into an integral over real variables, to make all components of the gauge fields $a_\mu$ and $b_\mu$ real, at the price of introducing integer link variables $Q_i$ and $M_i$,

$$
\begin{aligned}
& Z = \sum_{\{Q_i\}} \sum_{\{M_i\}} \int \mathcal{D}a_\mu \mathcal{D}b_\mu \int_{-\pi}^{+\pi} \mathcal{D}\varphi \, \exp(-S) , \\
& S = \sum_x i2\pi \, a_\mu K_{\mu\nu} b_\nu + \frac{1}{2\ell_0 E_J} j_i^2 + \frac{\pi^2}{4\ell_0 E_C} \phi_i^2 + i2\pi a_i Q_i + i2\pi b_i M_i \\
& + b_i \left(\hat{K}_{i0}\Delta_0\varphi + \hat{K}_{ij}\Delta_j\varphi\right) + b_0 \hat{K}_{0i}\Delta_i\varphi .
\end{aligned}
\tag{23}
$$

Finally. we note that the quantities $(1/2\pi)\hat{K}_{\mu\nu}\Delta_\nu\varphi$ are the circulations of the array phases around the plaquettes orthogonal to the direction $\mu$ in 3D Euclidean space-time and are thus quantized as $2\pi$ integers. We can thus absorb the quantities $\left(\hat{K}_{i0}\Delta_0\varphi + \hat{K}_{ij}\Delta_j\varphi\right)$ in a redefinition of the integers $M_i$ and define $\hat{K}_{0i}\Delta_i\varphi = 2\pi M_0$. The original integral over the phases $\varphi$ can then be traded for a sum over the vortex numbers $M_0$,

$$
\begin{aligned}
& Z = \sum_{\{Q_i\}} \sum_{\{M_\mu\}} \int \mathcal{D}a_\mu \mathcal{D}b_\mu \, \exp(-S) , \\
& S = \sum_x i2\pi \, a_\mu K_{\mu\nu} b_\nu + \frac{1}{2\ell_0 E_J} j_i^2 + \frac{\pi^2}{4\ell_0 E_C} \phi_i^2 + i2\pi a_i Q_i + i2\pi b_\mu M_\mu ,
\end{aligned}
\tag{24}
$$

which is the gauge theory partition function in the main text.

### 4.4. Summing over the Monopole Instanton Plasma

We start from the instanton plasma representation of the Wilson loop expectation value,

$$\langle W(C)\rangle = \frac{1}{Z_m} \sum_{\{m\}} e^{-\frac{\pi g}{\ell_0 \omega_P} \sum_x m_x \frac{1}{-\Delta} m_x} e^{i2\pi q_{ext} \sum_S \hat{\Delta}_i S_i \frac{1}{-\Delta_2} m_x} , \tag{25}$$

where $\Delta_2$ is the 2D Laplacian. Following Polyakov, we introduce a scalar field $\chi$ and we rewrite this as

$$\langle W(C)\rangle = \frac{1}{Z_{m,\chi}} \int \mathcal{D}\chi \, e^{-\frac{\ell_0 \omega_P}{4\pi g} \sum_x \Delta_i \chi \Delta_i \chi} \sum_N \frac{z^N}{N!} \sum_{x_1,\ldots,x_N} \sum_{m_1,\ldots,m_n = \pm 1} e^{i \sum_x m(\chi + q_{ext}\eta)} , \tag{26}$$

where the angle $\eta = 2\pi\hat{\Delta}_i S_i / (-\Delta_2)$ represents a dipole sheet on the Wilson surface $S$ and the monopole fugacity $z$ is determined by the self-interaction as

$$z = e^{-\frac{\pi g}{\ell_0 \omega_P} G(0)} , \tag{27}$$

with $G(0)$ being the inverse of the infrared-regularized 2D Laplacian at coinciding arguments. In (27) we also adopt the dilute instanton approximation, valid at low $g$, in which one takes into account only single monopoles $m = \pm 1$. The sum can now be explicitly performed, with the result,

$$\langle W(C)\rangle = \frac{1}{Z_\chi} \int \mathcal{D}\chi \, e^{-\frac{\ell_0 \omega_P}{4\pi g} \sum_x \Delta_i \chi \Delta_i \chi + \frac{8\pi g}{\ell_0 \omega_P} z(1 - \cos(\chi + q_{ext}\eta))} , \tag{28}$$

By shifting the field $\chi$ by $-q_{\text{ext}}$ and introducing $\mu^2 = 4\pi g z / \ell_0 \omega_P$, we can rewrite this as

$$\langle W(C) \rangle = \frac{1}{Z_\chi} \int \mathcal{D}\chi \, e^{-\frac{\ell_0 \omega_P}{2\pi g} \sum_x \frac{1}{2} \Delta_i (\chi - q_{\text{ext}} \eta) \Delta_i (\chi - q_{\text{ext}} \eta) + \mu^2 (1 - \cos(\chi))} \,. \tag{29}$$

For small $g$, this integral is dominated by the classical solution to the equation of motion

$$\Delta_2 \chi_{\text{cl}} = q_{\text{ext}} \Delta_2 \eta + \mu^2 \sin \chi_{\text{cl}} \,, \tag{30}$$

where $\Delta_2$ is the 2D spatial Laplacian. Far from the boundaries of the Wilson surface $S$, where $S_i \neq 0$, this reduces to a one-dimensional equation. Taking the Wilson surface $S$ to lie in the $(t, x_2)$ plane, this gives,

$$\hat{\Delta}_{x1} \Delta_{x1} \chi_{\text{cl}} = -2\pi q_{\text{ext}} \hat{\Delta}_{x1} S_1 + \mu^2 \sin \chi_{\text{cl}} \,. \tag{31}$$

Following [18], we solve this equation in the continuum limit,

$$\partial_{x1} \partial_{x1} \chi_{\text{cl}} = 2\pi q_{\text{ext}} \delta'(x_1) + \mu^2 \sin \chi_{\text{cl}} \,. \tag{32}$$

For $q_{\text{ext}} = 1$ (corresponding to Cooper pairs in our case), the classical solution with the boundary conditions $\chi_{\text{cl}} \to 0$ for $|x_1| \to \infty$ is

$$\chi_{\text{cl}} = \text{sign}(x_1) \, 4 \arctan e^{-\mu |x_1|} \,. \tag{33}$$

Inserting this back into (29), we get Formula (8) in the main text. The solution with the same boundary conditions for $q_{\text{ext}} = 1/2$ (corresponding to single electrons in our case), instead, is

$$\chi_{\text{cl}} = \theta(x_1) \, 4 \arctan e^{-\mu x_1} \,. \tag{34}$$

The action of this solution leads to Formula (10) in the main text.

**Author Contributions:** M.C.D., C.T., and V.M.V. conceived the work and carried out the calculations; N.P. and F.S.N. discussed the results; and all authors contributed to writing the manuscript. All authors have read and agreed to the published version of the manuscript.

**Funding:** The work at Argonne (V.M.V.) was supported by the U.S. Department of Energy, Office of Science, Basic Energy Sciences, Materials Sciences and Engineering Division.

**Acknowledgments:** M.C.D. thanks CERN, where she completed this work, for kind hospitality.

**Conflicts of Interest:** The authors declare no conflict of interest.

**Data Availability:** Data sharing not applicable to this article as no datasets were generated or analyzed during the current study.

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
