# Peer review of "Magnetic Monopoles and Superinsulation in Josephson Junction Arrays"

_quantumrep, doi:10.3390/quantum2030027_

Round 1

Reviewer 1 Report

In this manuscript the authors study the transport properties of the 2D Josephson arrays. Their goal is to suggest a mechanism of the transition to the state which they call "superinsulating state" which should be characterized by a very high resistance. They conclude that this transition can be understood by introducing magnetic monopoles as the objects in the (xyt) space which corresponds to the creation and further disappearance of the vortex in some unite cell of the array. They argue that this object is similar to the vortices in the (xt) space introduced earlier for the study of the phase slip centers in one dimensional superconductors. Before I could recommend the work for publication I would ask the authors to answer the following questions.

  1. I can admit of course that the description based on these monopoles is convenient for the authors but if I understand their explanations correctly the overall physical picture can be explained even without monopoles using only vortices. If so then I would like to ask the authors how the vortex needed for the formation of the monopole can appear. The point is that due to topology vortices in superconducting systems can be created either by their entry from the system edge or in the process of the creation of the vortex-antivortex pair. Another way is that the vortex in a given plaquette can disappear moving to the neighboring plaquette. The singly quantized vortex considered by the authors can not be split in parts because of the topology restrictions. What particular scenario consistent with these topological restrictions is considered by the authors? Most probably they just mean the vortex motion in the array. If so then the origin of the dissipation should relate to this vortex dynamics. Could the authors explain the origin of the superinsulating state using a standard language of the vortex dynamics (classical or quantum)? If it is possible it would be nice to see the estimates for the vortex tunneling probability, thermal activation and other quantities related to the transport characteristics.

2. If the author disagree with the above suggestion to use usual vortices instead of their monopoles for the understanding of the physics of these systems or probably they can suggest the way to overcome the above topological restrictions could they propose some experimental setup which could confirm the existence of monopoles in the Josephson arrays? 

Author Response

Reviewer #1

In this manuscript the authors study the transport properties of the 2D Josephson arrays. Their goal is to suggest a mechanism of the transition to the state which they call "superinsulating state" which should be characterized by a very high resistance. They conclude that this transition can be understood by introducing magnetic monopoles as the objects in the (xyt) space which corresponds to the creation and further disappearance of the vortex in some unit cell of the array. They argue that this object is similar to the vortices in the (xt) space introduced earlier for the study of the phase slip centers in one dimensional superconductors. Before I could recommend the work for publication, I would ask the authors to answer the following questions.

Answer:

We thank Reviewer #1 for a nice account of the results of our work. Before proceeding to the Reviewer’s detailed comments, we would like to complete the account of what is done in our work. The aim of our work is not to study transport in 2D Josephson junction arrays (JJA). This has been already done in numerous comprehensive studies, see Refs. [16,33,35] for review. Neither our goal is “to suggest a mechanism of the transition to the state which” we “call "superinsulating state".” The superinsulating state was suggested by Nobel Laureate Gerard ‘t Hooft as a state in which the quark matter resides in hadrons as early as in 1978 and was discovered to be realized in JJA by two authors of the present manuscript in 1996, see Ref. [1]. The idea that the formation of the superinsulating state is inherently related to Dirac’s monopoles was put forth in the preceding work of three authors of the present manuscript, Ref. [7]. The goal of the present work is to rigorously derive the origin of the monopole plasma that ensures the remarkable properties of the superinsulating state of matter. The Josephson junction array is an exemplary model system that has already allowed for the discovery of superinsulation. That is why we have chosen the JJA as a system that enables us to reveal rigorously the origin and the underlying mechanism for the origin of the monopole plasma.

  1. Reviewer #1

I can admit of course that the description based on these monopoles is convenient for the authors but if I understand their explanations correctly the overall physical picture can be explained even without monopoles using only vortices. If so then I would like to ask the authors how the vortex needed for the formation of the monopole can appear. The point is that due to topology vortices in superconducting systems can be created either by their entry from the system edge or in the process of the creation of the vortex-antivortex pair. Another way is that the vortex in a given plaquette can disappear moving to the neighboring plaquette. The singly quantized vortex considered by the authors cannot be split in parts because of the topology restrictions. What particular scenario consistent with these topological restrictions is considered by the authors? Most probably they just mean the vortex motion in the array. If so then the origin of the dissipation should relate to this vortex dynamics. Could the authors explain the origin of the superinsulating state using a standard language of the vortex dynamics (classical or quantum)? If it is possible it would be nice to see the estimates for the vortex tunneling probability, thermal activation and other quantities related to the transport characteristics.

Answer:

We would like to stress from the very beginning that vortices, in their classical sense, do not lead to the existence of superinsulation. Using monopoles is not a question of the authors’ taste or convenience, but it is unavoidable. They are simply the correct mechanism explaining superinsulation. The goal of our work is to present the mechanism for the formation of superinsulation in JJA. We demonstrate rigorously that this is the formation of the monopole plasma. Vortices are not sufficient.

To explain why it is impossible to provide superinsulation by vortices, we have to invoke the physics of instantons, for which an excellent review can be found in Sydney Coleman’s famous Erice lectures “The uses of instantons”, in his book “Aspects of symmetry” published by Cambridge University Press, or, alternatively in Polyakov’s book “Gauge Fields and Strings”, Ref. [18] in the manuscript. On a crude qualitative level, we note that in order to squeeze  Faraday’s electric field lines into electric strings that ensure linear charge confinement, one needs what can be called “phase Bose condensate” presuming the non-dissipative phase dynamics. Since vortices, as the Reviewer noticed her/himself have to move with dissipation, vortices cannot provide the mechanism for superinsulation.

Topological solitons, a prominent example of which are vortices in 2D gauge theories and Josephson junction arrays, are indeed characterized by a topological quantum number, which in this case is the winding number of the phase around a plaquette, as pointed out by the referee. These quantum numbers are called topological since their values cannot be changed by local deformations of the fields/phases but require a global change over the whole system/array. Such a change involves high energy barriers, that is why topological quantum numbers are often referred to as topologically protected numbers.

However, topological quantum numbers, contrary to Noether quantum numbers, are not always conserved (see the above literature)! There are non-perturbative finite action field configurations in Euclidean space-time which represent tunneling events in real space-time so that the above energy barrier is tunneled through. These finite-action solutions in the Euclidean space-time are called instantons. Of course, these instantons must carry exactly one unit of the topological quantum number that they “create” or “destroy”. And because this quantum number involves globally the degrees of freedom of the whole system/array, of course, also the instanton, albeit having a localized core, carries long-range fields involving all the global degrees of freedom of the system/array.

In the present case, the instanton is a magnetic monopole in the Euclidean three-dimensional space and, as required, it carries exactly one unit of the topological quantum number and long-range fields that extend over all the array. Therefore, all the plaquettes of the array are involved in one single tunneling event!

The Reviewer is absolutely right in saying that a single phase slip of 2\pi corresponds to a vortex moving from one plaquette to the other. However, these local phase slips are not sufficient for creating superinsulation! Instead, monopoles are exactly the sought global events involving all the plaquettes of the array and cannot be described in terms of local variables, like the local phases in the corner of one single plaquette. Monopoles are more complex configurations in which one vortex can locally appear or disappear in a tunneling event and the dynamics of which cannot be reduced to the dynamics of vortices themselves. The only requirement is that overall magnetic flux is conserved in Euclidean space-time and this is exactly what is depicted in Fig. 1. in which a 2\pi vortex on the lower plaquette disappears. The overall flux conservation is guaranteed by the fact that fluxes \pi/2 exit the 4 vertical plaquettes and so we have a single magnetic monopole in the center. Note that these fluxes exiting the vertical plaquettes continue to infinity (or the sample boundaries) to guarantee overall flux conservation and this is why, in the end, all plaquettes of the array are involved in this event. Again, using a graphic language, the whole picture is similar to the quantum mechanical description of a particle. There are many situations where the particle cannot be viewed as a solid ball but should be represented as a multicomponent and delocalized wave function. Therefore, the reason why it is not possible to explain the phenomenon of superinsulation within the vortex dynamics picture is exactly the same way it is not possible to explain superconductivity within the framework of Newtonian mechanics of ball-like particles.

  1. Reviewer #1

If the author disagree with the above suggestion to use usual vortices instead of their monopoles for the understanding of the physics of these systems or probably they can suggest the way to overcome the above topological restrictions could they propose some experimental setup which could confirm the existence of monopoles in the Josephson arrays? 

Answer:

We thank the Reviewer for this nice and important question. This question has several interesting and appealing aspects. There have been recently developed devices that enabled researchers to identify single monopoles in condensed matter systems, see, for example, A. Uri et al. Nanoscale imaging of equilibrium quantum Hall edge currents and of the magnetic monopole response in graphene, Nature Physics, 16, 164 (2020). Using the same SQUID-on-tip, a nanoscale superconducting quantum interference device (SQUID) fabricated on the apex of a sharp pipette, which is now possible to use at temperature 300 mK. A possibility of going below 100 mK, which is necessary in order to deal with superinsulation in TiN and NbTiN films, where it was conclusively established experimentally, is being discussed with this group. This is still a forthcoming project since some technical challenges related to going down in temperature still remain. Therefore, for the moment the best route for catching monopoles is to explore the consequences of having a monopole plasma (exactly the same way as investigating the implications of having a Cooper pair condensate in superconductors), which includes, firstly, the detection of the superinsulating state in JJA and then a careful study of the I-V curves that are characteristic to superinsulation. In particular, the monopole mechanism implies a linear binding potential between charges and holes which is characterized by a typical length scale. When samples are larger than this typical scale one should measure hyperactivation, when samples are smaller than this typical scale one should measure metallic saturation of the resistance because charges are essentially free below this scale. This size-dependence has been already measured in superinsualting thin films of NbTiN (ref. [13] in the manuscript) and its realization in JJA would become a direct confirmation of the presence of the plasma of magnetic monopoles.

Reviewer 2 Report

In this manuscript, the authors studied the superinsulation phase in a two-dimensional Josephson junction array. The magnetic monopoles are obtained as quantum instantons tunneling. The authors show that the logarithm of the Wilson loop is proportional to the area of the surface S enclosed by a loop in 3d Euclidean space-time. The results are useful to understand the quark confinement and to the possible experimental realization of a superinsulate state in condensed matter systems. The manuscript is well written and its exhaustive in the mathematical details, although I have to read several references in order to understand critical steps, mostly in the  4.4 demonstration. I consider this work suitable for publication, although I have some doubts about C_0=0 limit: as the authors claims, to detect superinsulation in JJA implies small interdistance between centres of the superconducting islands, then the interlayer coupling E_j will be sensitive to this distance and will increases when the JJA-interdistance decreases. In turn, the interlayer coupling E_j should renormalize the mass of the charges, where the mass is written in terms of the mutual capacitances and the capacitances of the islands to the ground. This would imply that is not trivial to take the C_0=0 limit and in turn, different values for the string size. Once this is clarified the manuscript will be suitable for publication.

Author Response

File is attached

Round 2

Reviewer 1 Report

The reply of the authors to my questions are rather detailed and they indeed clarify the key points of the paper. The authors believe that the phenomena considered in their work can not be understood within the vortex-like picture and all the plaquettes of the array are involved in one single tunneling event and formation of the monopole. If so I would expect the infinitely small probability of this tunneling event which seems to be in contradiction with the opinion of the authors. Still after some hesitations I decided to recommend the paper for publication since it contains some new exciting suggestions and  can stimulate further interesting discussions in the field. At the same time I recommend the authors to include optionally more discussions of their statement that the phenomena studied in their work can not be understood within the standard vortex physical picture (even involving quantum tunneling of vortices).